# Charging dynamics of an individual nanopore

Ran Tivony[1], Sam Safran[1], Philip Pincus[2], Gilad Silbert[1,3] & Jacob Klein [1]

Meso-porous electrodes (pore width « 1 μm) are a central component in electrochemical energy storage devices and related technologies, based on the capacitive nature of electric double-layers at their surfaces. This requires that such charging, limited by ion transport within the pores, is attained over the device operation time. Here we measure directly electric double layer charging within individual nano-slits, formed between gold and mica surfaces in a surface force balance, by monitoring transient surface forces in response to an applied electric potential. We find that the nano-slit charging time is of order 1 s (far slower than the time of order $3 \times 10^{-2}$ s characteristic of charging an unconfined surface in our configuration), increasing at smaller slit thickness, and decreasing with solution ion concentration. The results enable us to examine critically the nanopore charging dynamics, and indicate how to probe such charging in different conditions and aqueous environments.

[1] Department of Materials and Interfaces, Weizmann Institute of Science, Rehovot 76100, Israel. [2] Physics Department, University of California, Santa Barbara,, CA 93106, USA. [3]Present address: Adama Makhteshim Ltd, Beer Sheva 84100, Israel. Correspondence and requests for materials should be addressed to J.K. (email: Jacob.klein@weizmann.ac.il)

Conducting materials with a large surface-to-volume ratio, such as meso-porous electrodes (pore widths « 1 μm), are important in a range of technologies[1-6]. Prominent among these are supercapacitors[4,5], but also batteries[7,8], fuel cells[9,10] and electrocatalysts[11,12], while emerging applications include capacitive deionization[13,14], and the extraction of renewable energy through capacitive mixing of aqueous solutions of different salt concentrations[15,16]. Charge is stored in supercapacitors when electrolyte ions form (capacitive) electrical double layers (EDLs) at the surface of oppositely charged electrodes under an externally applied voltage[17]. Since the amount of charge stored is proportional to the available electrode surface area, materials with a high specific area (such as porous electrodes) are clearly at a premium; but they are only fully useful if ion transport enables equilibrium EDL formation over the entire pore within the relevant charging/discharging time. It is crucial therefore to be able to characterize and understand the dynamics of EDL formation within nano-pores.

In equilibrium, the EDL at a charged surface screens the associated electric field, with an exponential decay length (the Debye length) $\lambda_D = \sqrt{\frac{\varepsilon\varepsilon_0 k_B T}{2c_0 e^2}}$ (for 1:1 electrolyte), where $c_0$ is the concentration of salt in the bulk solution, $\varepsilon_0$ is the permittivity of free space, $\varepsilon$ is the dielectric constant, $k_B T$ the thermal energy, and $e$ the electronic charge[18]. Changes in the surface potential rearrange the ionic atmosphere comprising the EDL to a new equilibrium configuration[19], a dynamic process known as EDL charging. This is well understood for planar electrodes in bulk solution, where the characteristic charging time is defined by $\tau_c = \lambda_D H / \mathcal{D}$ (where $H$ is the distance between electrodes and $\mathcal{D}$ is the ion diffusion coefficient)[19-22], but for the case of EDL charging in nano-confined geometries (as in porous electrodes) the dynamics may be much slower. This is because, in small diameter pores, changes to the EDL are constrained by the limited supply of ions within the pore[23-27] and the time for ions to access the entire pore from the reservoir with which it is in contact.

A widely used description of ion transport in pores is the classic "Transmission-Line" (TL) model, by de Levie[6,24,28], where the pore – a closed cylinder of length $L_p$ and diameter $h_p$ ($L_p$ » $h_p$ » $\lambda_D$) in contact with an ion reservoir - is treated as an equivalent TL circuit composed of a set of resistors and capacitors, yielding a time for EDL charging within the pore $\tau_{TL} = (L_p^2 / \mathcal{D})(\lambda_D / h_p)$. This characteristic time scales as the diffusion time ($L_p^2/\mathcal{D}$) for ions to traverse the pore length $L_p$, modulated by the ratio ($\lambda_D/h_p$). The latter is the extent of the near-surface EDL region (~$\lambda_D$), within which ion redistribution must occur due to the potential change, relative to the pore diameter $h_p$, within which little overall change occurs. The charging dynamics of an EDL confined in a single nano-pore have been investigated primarily theoretically, e.g. refs. [6,29-35], while several treatments have extended the TL treatment, and different charging dynamics have recently been predicted for EDLs confined in dimensions comparable with its Debye length[6,28,36]. Experimentally, charging dynamics have been probed within (macroscopic) porous electrodes[25,37-39], but, to our knowledge, no measurements of EDL charging dynamics within a single nano-pore – the basic element of such electrodes—have been reported to date.

Here we use a surface force balance (SFB) to probe experimentally the charging dynamics of an EDL confined to a nano-slit between two surfaces, in response to a step-change in the surface potential (or charge) of one of them (arising from a step $\Delta\Psi_{app}$ in the potential applied to it). We find that the response to such a step is governed by two successive processes: Immediate formation of a strong electric field across the pore due to elimination of the screening, arising from the instantaneous charge imbalance in the EDL, followed by a slower process in which the ionic atmosphere of the EDL rebuilds to screen the generated electric field while charging the nano-pore interior (the gold acquires its new surface potential value over a time $\delta t$ much faster than EDL charging time as it is associated with drift motion of electrons which is orders of magnitude faster than for ions in solution[11,40]). Our results show that full EDL charging within the nano-slit occurs over times of order 0.5–1 s, far longer than $\tau_c \approx 3 \times 10^{-2}$ s in our configuration (where $\tau_c = \lambda_D H / \mathcal{D}$, and the values of $\lambda_D$, $H$ and $\mathcal{D}$ are given below). They reveal that the charging time increases at smaller slit thickness, and decreases with increasing ion concentration in the electrolyte solution, suggesting that the process of EDL charging within the nano-slit is largely limited by diffusion of ions from the bulk reservoir to which the nano-slit is coupled[24,28,36]. These findings demonstrate the ability of the SFB to access and probe the EDL charging dynamics within a single nanoscale gap (« 1 μm), the basic element of porous electrodes, providing a method for probing EDL charging in nano-confined slits with different surfaces and under different conditions.

## Results

**Probing EDL charging in a nano-slit with the SFB**. Figure 1a shows a schematic of the experimental SFB configuration used in this study, where a single-crystal mica surface faces a smooth gold surface (r.m.s. roughness ca. 3 Å) at a controlled potential[41-43]. The absolute separation D between the surfaces is determined (to ± 3 Å) by multiple beam interference and monitored via video-recording of the wavelengths of the interference fringes[44,45].

We probe the perturbation and the charging of the EDL through its transient effect on the force between the gold and mica surfaces. A typical dynamic measurement, conducted in 5 mM NaNO₃ ($\lambda_D = 4.3$ nm), is shown in Fig. 1c. Initially, the surfaces are brought to a given initial separation $D_i$ (» $\lambda_D$). Then, either a positive ($-0.2$ V $\rightarrow +0.2$ V) or negative ($+0.2$ V $\rightarrow -0.2$ V) potential step $\Delta\Psi_{app}$ is applied to the gold while monitoring the gold-mica separation D (Fig. 1c). After $\Delta\Psi_{app}$ is applied at time $t = 0$ (Fig. 1d), the gold electrode – both outside and within the confined nano-slit – accumulates a new surface charge density over a very short time $\delta t$ (typically < $10^{-9}$ s due to drift motion of electrons[11,40]) as it attains its new surface potential value $\Psi_{gold}$. This generates (at $\delta t$) an unscreened electric field[19] which exerts an electrostatic force $F_e(t = \delta t \approx 0)$ on the negatively charged mica, bending the spring $K_n$ on which it is mounted by $\Delta D$ to an extremal separation $D_f$ relative to the gold, over a time $\Delta t_s$. As ions transport to reconstruct the EDL (EDL charging) within the nanopore, $F_e$ is progressively screened and the surfaces relax to their initial separation $D_i$ over a further time $\Delta t_r$ (Fig. 1d), since, in all cases in our study, $D_i$ » $\lambda_D$. This transient motion manifests as a peak, with a characteristic asymmetric shape, in the D vs. t trace (Fig. 1d) and a period of ($\Delta t_s + \Delta t_r$) which characterizes the charging time of the EDL within the nano-slit. Away from the nano-slit – at the unconfined gold surface - the EDL equilibrates over a time $\tau_c$[19] ( $= \lambda_D H / \mathcal{D} \approx 3 \times 10^{-2}$s), which is much shorter than the EDL charging time (0.5–2 s) within the nanopore measured in our experiments. For example, for the configuration of Fig. 1d, a characteristic time ($\Delta t_s + \Delta t_r$) ≈ 0.85 s was obtained in response to a potential step of -0.2 V → + 0.2 V. Overall, this indicates three different time scales following the potential step: ($\Delta t_s + \Delta t_r$) ≫ $\tau_c$ ≫ $\delta t$. Thus, the times ($\Delta t_s + \Delta t_r$) we measure in our experiments, corresponding to the EDL equilibration within the nano-pore, are well separated from $\tau_c$, the time for EDL equilibration at the unconfined gold away from the nanopore.

The initial ($t = 0$) instantaneous electric field across the gap depends on the effective surface potentials $\Psi_{gold}$ at the gold surface at different applied potentials $\Psi_{app}$; these may be extracted from normal force vs. surface-separation D profiles, as shown in

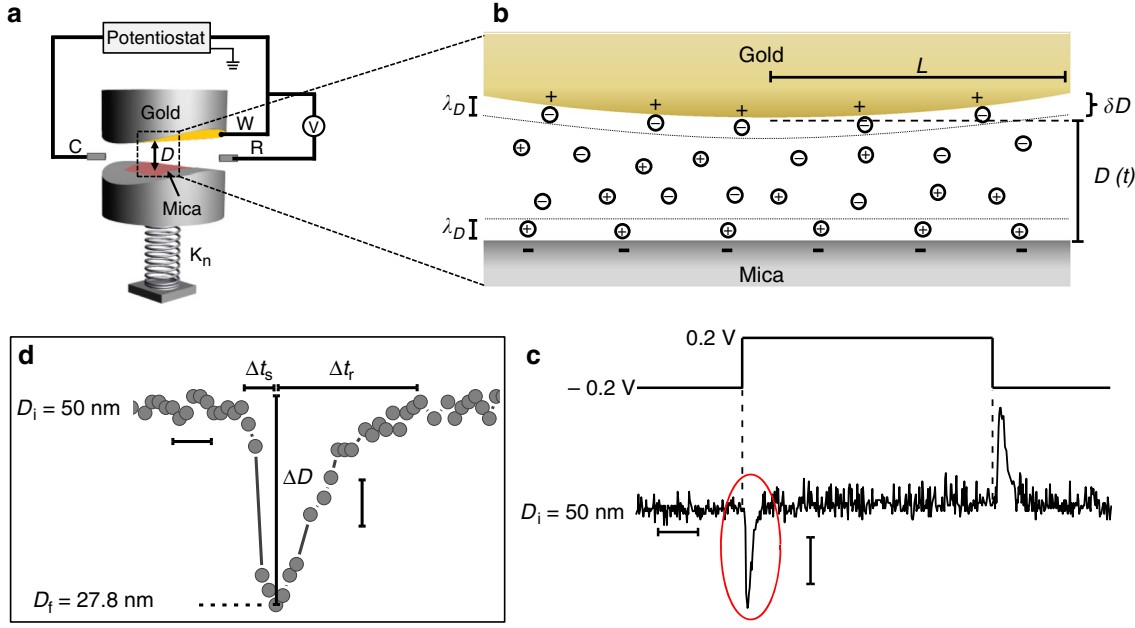

**Fig. 1 a** Schematic of the Surface Force Balance (SFB) and three-electrode configuration, with the principle components labeled, as described in detail in the experimental section. During measurements, the electrodes are immersed in an electrolyte solution inside a quartz bath, custom-designed to prevent leakage of current to the ground. Potentials applied to the gold surface were in the so-called double-layer range (i.e., where the electrode is ideally polarized), to ensure, crucially, the absence of any electrochemical reactions[44,45] (Supplementary note 5). **b** A section through the intersurface gap at closest separation $D$, showing its pore-like structure, where $\lambda_D$ is the Debye length, $L$ is the radius of the circular pore, $\delta D$ is the change in pore width at a distance $L$ from the pore center due to curvature of the surfaces ($\delta D \cong L^2/2R$). The schematic is not to scale: $L$ is typically 100 μm, while $D$ is of order 100 nm, giving a (diameter/thickness) ratio of ca. 1000 for the pore/slit. **c** $D(t)$ traces taken in 5 mM NaNO$_3$ at $D = 50$ nm based on video recording of the motion of the interference fringes in the SFB, in response to positive and negative potential steps as indicated by the upper potential trace. The obtained peaks reflect the movement of the lower surface (mica) in response to application of a positive ($-0.2$ V $\rightarrow 0.2$ V) or negative ($0.2$ V $\rightarrow -0.2$ V) step potentials. Scale bars: horizontal – 2 s; vertical – 10 nm. **d** A magnified view of the peak circled in red in the $D(t)$ plot in (**c**), demonstrating its asymmetric shape, where $\Delta t_s$ and $\Delta t_r$ signify the initial motion and relaxation time, as shown, and $\Delta D$ is the distance shift from initial surface separation $D_i = 50$ nm to extremal separation $D_f = 27.8$ nm (prior to its relaxation back to $D_i$). Scale bars: horizontal - 0.2 s; vertical - 5 nm

Fig. 2a. In Fig. 2b are shown $D(t)$ traces in response to $\Psi_{app} = -0.2$ V $\rightarrow +0.2$ V and back again, at selected $D_i$ values covering the range of pore widths $D_i$ examined. Importantly, this also demonstrates the repeatability and reproducibility of our dynamic measurements, indicating that no chemical reaction occurs at the gold (this is shown directly by cyclic voltammetry, ref. [45] and in the Supplementary Figure 4). In Fig. 2c are magnified peaks at different $D_i$ values. Such transient peaks reflect the EDL perturbation by $\Delta\Psi_{app}$ (at $t = 0$) and its subsequent charging, and are analyzed below (for clarity we focus on the $-0.2$ V $\rightarrow +0.2$ V transition though identical considerations apply to the reverse transition).

**Transient forces during EDL charging.** The force $F_e(t)$ attracts the charged mica surface due to the induced positive charge on the gold surface, changing the surface separation $D_i$ by $\Delta D(t) = (D_i - D(t))$. This motion is opposed by the bending of the spring (of constant $K_n$) on which the mica surface is mounted, which exerts an opposing force $F_k = K_n\Delta D$, and by a hydrodynamic damping force $F_H$ (arising from extrusion of the liquid as the surfaces approach or separate). Van der Waals (vdW) attraction between the surfaces is negligible since $D$ is always > 20 nm, and we may set the normal surface force as equal to $F_e$. The equation of motion is thus $F_e(t) = F_k + F_H + m(d^2D/dt^2)$, where the last term is inertial and m is the mass of the mica surface and its mount. We note that the magnitude of $m(d^2D/dt^2)$ (where $m \approx 3 \times 10^{-3}$ kg), as determined from the $D$ vs. $t$ plots (Fig. 1d or

Fig. 2b), is ca. $10^{-9}$N, which is negligible compared with the hydrodynamic ($F_H = O(10^{-6}$ N)) and spring ($F_k = O(10^{-6}$ N)) force terms, and may be ignored. The hydrodynamic force, between a sphere approaching a flat a closest distance $D$ away across a liquid of viscosity $\eta$, is[46] $F_H = 6\pi R^2\eta[(dD/dt)/D]$, and the equation of motion becomes:

$$F_e(t) = K_n\Delta D(t) + 6\pi R^2\eta[(dD/dt)/D(t)] \qquad (1)$$

Before solving equation 1, we address an important qualitative point concerning the EDL charging. Following the potential change on the gold surface at $t = 0$, $F_e$ is a maximum but immediately begins to decay as the EDL recharges. During the initial motion ($t < \Delta t_s$) $F_e$ exceeds $F_k$, but at $t = \Delta t_s$, where $D = D_f$ (Fig. 1d), it has decayed so that $F_e(\Delta t_s) = F_k = K_n(D_i - D_f)$. At longer times $t > \Delta t_s$, $F_e$ becomes smaller than the spring force $F_k$, and the mica surface is pushed back towards its equilibrium position $D = D_i$ over the period $\Delta t_r$. The crucial point is whether the period $\Delta t_r$ is due to hydrodynamic damping alone, or to a slower process due to $F_e$ decay arising from increased screening associated with EDL charging limited by ion transport within the gap. To resolve this we solve for the motion of the surfaces from $D = D_f$ at the peak to $D = D_i$, on the assumption of hydrodynamic damping alone. The relevant equation is simply equation (1) with $F_e$ set to zero, and boundary condition $D = D_f$ at $t = 0$. The resulting $D(t)$ variation is given in Fig. 3 and shown as the red curves in Fig. 3a. We see that for traces at

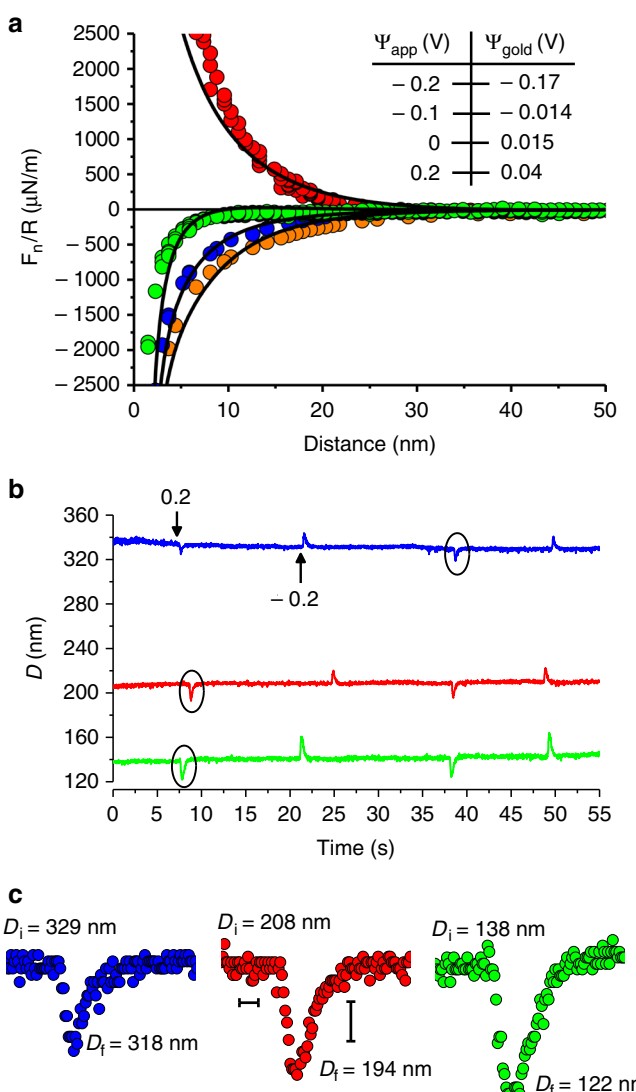

**Fig. 2 a** Interaction profiles $F_n(D)/R$ between gold and bare mica surfaces across 2 mM NaNO$_3$, at a given contact point, under different applied potentials $\Psi_{app}$. Gold surface potential $\Psi_{gold}$ values in legend are extracted from respective fits (black curves) to the PB equation with constant charge (mica) vs. constant potential (gold) boundary conditions, augmented by vdW attraction[44,45]. **b** An example for dynamic measurements ($D(t)$ plots) taken at different surface separations, in response to positive and negative step potential between − 0.2 V and + 0.2 V. The peaks indicate the transient movement of the lower surface (mica) in response to application of the step potential changes. Measurements both of the interaction profiles (**a**) and the response to potential changes were performed between the same surfaces and salt solution (2 mM NaNO$_3$). **c** A magnified view of transient peaks (circled in the $D(t)$ plots in **b**) obtained at different gold-mica separations, where $D_i$ and $D_f$ are the initial and extremal surface separation values, respectively. Scale bars: horizontal – 0.2 s; vertical – 5 nm

$D_i \gtrsim 80$ nm, $D$ approaches $D_i$ significantly slower than can be accounted for by hydrodynamic damping alone. This shows that $D(t)$ is dominated by the slow decay in $F_e$ rather than by hydrodynamic damping, so that $(\Delta t_s + \Delta t_r) = \tau_{EDL}$ provides a measure of the time $\tau_{EDL}$ for the EDL charging (we note that for both hydrodynamic damping and EDL charging, the characteristic times vary as (1/D); however, at small $D_i$ the mean value of D across the pore is significantly larger than $D_i$, which reduces the

EDL charging time but leaves the hydrodynamic damping time unchanged, so that it becomes more dominant at the smallest $D_i$ values such as $D_i < $ ca. 80 nm. This is also manifested in the $\Delta D$ vs. $D_i$ plot (Supplementary Figure 1) as a decrease in $\Delta D$ at $D < $ ca. 80 nm. To solve equation (1), we require an explicit form for $F_e(t)$. We approximate the instantaneous initial force (just after application of the potential change on the gold surface) as $F_e(0) = \pi R \varepsilon \varepsilon_0 (\Delta \Psi_{eff})^2 / D(t=0)$, which is the force between a conducting sphere (radius R) and a conducting plane a closest distance $D$ apart[47] ($D \ll R$) differing in potentials by $\Delta \Psi_{eff}$. In our configuration $\Delta \Psi_{eff} = \Psi_{gold,eff} - \Psi_{mica}$ (at $t=0$), the effective potential difference at $t=0$ between the opposing gold and mica surfaces, where $\Psi_{gold,eff}$ is the instantaneous effective gold surface potential ($t=0$) arising from the abrupt variation of the gold surface charge (Supplementary Note 1). This may be evaluated from the potentials extracted from the force vs. distance profile at the relevant applied potentials[44], Fig. 2a. The decay with time of $F_e(t)$ is complex, depending on flux of ions between the bulk and the nano-slit which in turn modifies the initial gold surface charge (i.e. at $t = \delta t$) within the confined area until the EDL is fully relaxed. To proceed, we may assume that $F_e$ decays exponentially with a characteristic time $\tau$ due to the progressive screening of the electrostatic force with time, as the near-surface ion concentration rearranges following the potential change. Equation (1) then becomes:

$$\left[\pi R \varepsilon \varepsilon_0 \left(\Delta \psi_{eff}\right)^2 / D(t)\right] e^{-t/\tau} = K_n \Delta D(t) + 6 \pi R^2 \eta [(dD/dt)/D(t)]$$

(2)

We seek an approximate solution by recognizing that $D$ does not change too greatly from $D_i$ during the transient response (particularly for the larger $D_i$ values), and replace $D$ by $D_i$ in the denominators of the terms for $F_e$ and $F_H$. This gives:

$$\left[\pi R \varepsilon \varepsilon_0 \left(\Delta \psi_{eff}\right)^2 / D_i\right] e^{-t/\tau} = K_n \Delta D(t) + 6 \pi R^2 \eta [(dD/dt)/D_i],$$

(2')

which for boundary condition $D = D_i$ at $t = 0$, solves as:

$$D(t) = D_i + \left[F_e(0) \tau e^{-t/\tau} (\exp(t(f_H - K_n \tau)/f_H \tau) - 1)\right]/(K_n \tau - f_H),$$

(3)

where $F_e(0) = \pi R \varepsilon \varepsilon_0 \Delta \Psi_0 (\Delta \Psi_{eff})^2 / D_i$ and $f_H = 6 \pi R^2 \eta / D_i$.

Equation (3) describes the transient change in $D$ in response to $\Delta \Psi_{app}$. We note that at the peak in $D(t)$, at $t = \Delta t_s$, $(dD/dt) = 0$ so that for this value of $t$ ($= \Delta t_s$), equation (2') yields the value of $\tau$ directly. This may then be used in the expression for $D$ (equation (3)). In Fig. 3b, this is done for the $D$ vs. $t$ traces at different $D_i$ values, showing a close quantitative fit with no adjustable parameters (all parameters are determined, Fig. 3). We confirmed the validity of our approximate solution by numerically solving equation (2) (Supplementary note 4). The numerical solution shows the behavior of $D(t)$ to be close to that predicted by equation (3) over the range of $D$ values in our experiments, and as expected, closest at higher $D$ values where the approximation leading to equation (3) is best (Supplementary Figure 3).

**Effect of confinement on EDL charging time.** In Fig. 4 we gather together results for the EDL charging times for different $D_i$ values, that is, different thicknesses of the nano-slit, showing the variation with $D_i$ of $\Delta t_s$, $\Delta t_r$ and ($\Delta t_s + \Delta t_r$) (inset to Fig. 4a). While the values shown are for $\Delta \Psi_{app} = -0.2$ V $\rightarrow + 0.2$ V and + 0.2 V $\rightarrow -0.2$ V in 2 mM salt, similar behavior is seen at different salt concentrations (Fig. 4b), and for different salts (Fig. 4c), as well as for other potential variations (Supplementary note 3). In Fig. 4a

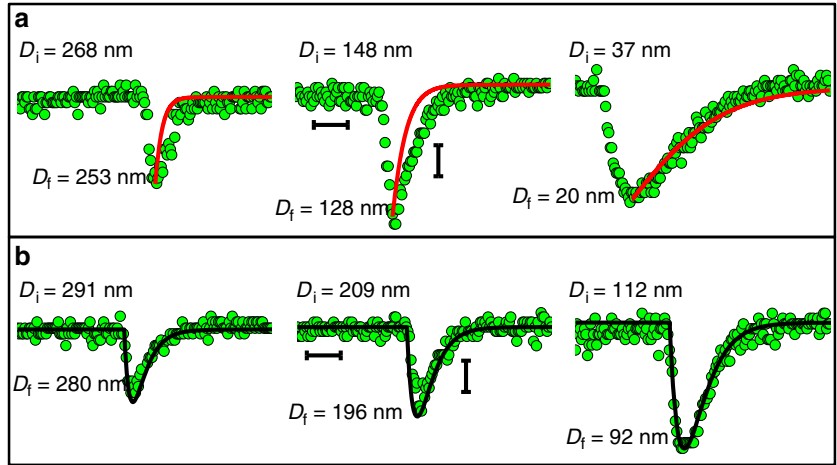

**Fig. 3** A magnified view of peaks measured at different surface separations ($D_i$) in 2 mM NaNO$_3$. **a** The red solid lines describe the motion of the lower surface from $D_i$ to $D_f$ in the presence of hydrodynamic damping alone as obtained by the analytical solution $D(t) = D_i D_f / \lceil \exp(-t k_n D_i / 6\pi R^2 \eta)(D_i - D_f) + D_f \rceil$ to the equation of motion $k\Delta D(t) = 6\pi R^2 \eta[(dD/dt)/D(t)]$ with a boundary condition $D(t = 0) = D_f$. They show that for the larger $D_i$ values ($D_i >$ ca. 80 nm, see Fig. 4a) the relaxation of the surfaces from $D_f$ back to $D_i$ is significantly slower than can be accounted for by hydrodynamic damping. Scale bars: horizontal – 0.4 sec; vertical – 5 nm. **b** The black solid lines are the solution to equation (3) which describes the transient motion in surface separation $D(t)$ in response to a positive step potential (– 0.2 V → + 0.2 V), where for all traces the parameters are fully determined (no adjustable parameters) (Supplementary note 6). Scale bars: horizontal – 0.4 sec; vertical – 5 nm

the broken and solid black curves are the calculated values of $\Delta t_s$ and $\Delta t_r$, respectively, from equation (3). The red curve in Fig. 4a is the calculated hydrodynamic damping time from the peak maximum (corresponding to the red curves shown in Fig. 3a), in the absence of any electrostatic force $F_e$, and shows that such damping alone cannot account for the significantly longer decay times $\Delta t_r$ at $D_i \gtrsim 80$ nm.

Figure 4b shows the variation of the overall charging time (($\Delta t_s(D_i) + \Delta t_r(D_i)$)) with $D_i$ for two different salt concentrations, 2 mM (squares) and 5 mM (triangles), where the solid black curves are the calculated fits from equation (3) for the two cases. These curves show good agreement with the respective measured charging times ($\Delta t_s(D_i) + \Delta t_r(D_i)$), indicating that our equations, which provide a description of the dynamics, though they do not clarify the mechanism of the nano-slit charging, can provide a good description of the data. The blue curve is the variation of the characteristic charging time $\tau_{TL}$ according to the transmission line model for nano-pore charging, described earlier and discussed in the following section.

Figure 4c shows the variation of the overall charging time when the NaNO$_3$ salt in Fig. 4a, b is replaced by a different salt, LiClO$_4$, which moreover has much larger differences between its ionic diffusion coefficients (Li$^+$ and ClO$_4^-$), as considered in the following section.

**Mechanism of nano-slit charging**. It is of interest to consider the EDL charging dynamics in the context of the classic transmission line (TL) approach noted earlier[24,28,36]. In our configuration (Fig. 1b), the nano-pore or nano-slit is the gold-mica gap about the region of closest approach, $D = D_i$, strictly the region between two crossed cylinders of mean radius $R$. Its effective geometry – as in the Derjaguin approximation[18], which is valid for $D_i \ll R$ in the present case – is that of a circular slit bounded by a spherical surface and a flat, with a slit radius $r = L$, in contact with an ion reservoir (because the slit is formed by two orthogonal cylindrical surfaces, it is symmetric about its midplane). $L$ is the distance required by ions from the reservoir (at $r = L$) to fully permeate the pore to its center ($r = 0$), where its thickness is $h_p = D_i$, to

compensate for the charge induced on the gold by the potential step $\Delta\Psi_{app}$. Within the TL model, the characteristic charging time of the EDL within such a circular slit is then expected to scale as $\tau_{TL} = (L^2 / \mathcal{D})(\lambda_D/h_p)$[6,24]. The surface separation at $r = L$, $D = D_i + (L^2/2 R)$ (Fig. 1b), must be sufficiently large to provide for the required excess of ions. This corresponds to a value $L \approx (2 R\Delta\sigma / c_0 e)^{1/2}$ (Supplementary note 2), where $\Delta\sigma$ is the change in the gold surface charge density $\sigma$ arising from $\Delta\Psi_{app}$. In Fig. 4b, it is shown as a solid blue curve, the characteristic time $\tau_{TL} = (L^2 / \mathcal{D})(\lambda_D / h_p)$ expected for a pore of length $L$, and width $h_p = D_i$. The fit to the experimental data corresponds to $L = 150$ μm. This is within a factor 2 of the value $L = (2 R\Delta\sigma / c_0 e)^{1/2} \approx 87$ μm evaluated for the corresponding potential step ($\Delta\Psi_{app} = + 0.2$ V → –0.2 V in 2 mM salt) (Supplementary Note 2), which is believed to arise from the different geometries considered in the two cases.

The characteristic time $\tau_{TL} = (L^2 / \mathcal{D})(\lambda_D / h_p)$ also predicts that EDL charging is faster for higher salt concentrations, since (in our configuration) a larger $c_0$ is associated both with a smaller $L$ ($\sim c_0^{-1/2}$), as well as a smaller $\lambda_D$ ($\sim c_0^{-1/2}$), suggesting $\tau_{TL} \sim c_0^{-3/2}$. This is indeed observed for EDL charging from higher salt concentrations as shown in Fig. 4b, where the experimental charging time is roughly 3.5-fold larger in the 2 mM NaNO$_3$ salt compared with the 5 mM salt, similar to the expected ratio ($c_{0,\ 2\,mM}/c_{0,\ 5\,mM})^{-3/2} \approx 4$.

In addition, we would expect from the expression for $\tau_{TL}$ that ions with different diffusion coefficients $\mathcal{D}$ would result in different charging times. Additional cations need to permeate the slit from the reservoir when the applied potential change renders the surface charge more negative (e.g. $\Delta\Psi_{app} = + 0.2$ V → –0.2 V), and additional anions need to permeate for the opposite case. For the NaNO$_3$ salt, Fig. 4a, b, where the $\mathcal{D}$ values for Na$^+$ ($\mathcal{D} = 1.3 \times 10^{-9}$ m$^2$ s$^{-1}$) and NO$_3^-$ ($\mathcal{D} = 1.9 \times 10^{-9}$ m$^2$ s$^{-1}$) differ by some 40%, there is a slight indication that EDL charging is slower when $\Delta\Psi_{app} = + 0.2$ V → – 0.2 V (red squares in Fig. 4a, when it is controlled by transport of slower Na$^+$ ions) relative to $\Delta\Psi_{app} = – 0.2$ V → + 0.2 V (green squares, when charging occurs by transport of NO$_3^-$ ions). However, when NaNO$_3$ is replaced by LiClO$_4$, where the $\mathcal{D}$ values for the latter ($1 \times 10^{-9}$ m$^2$ s$^{-1}$ and

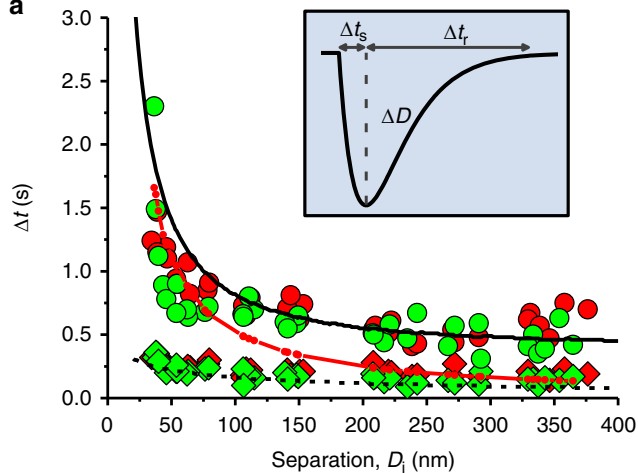

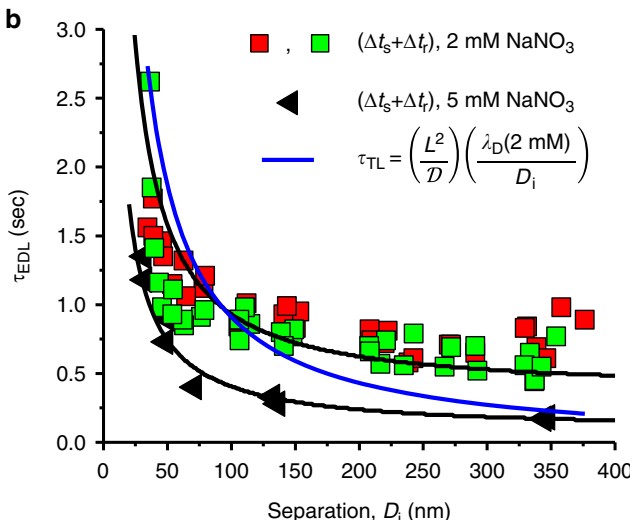

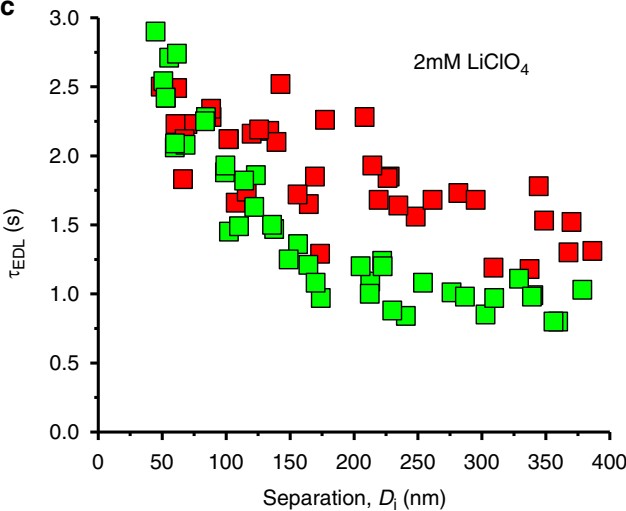

**Fig. 4** EDL charging at different nano-slit thickness $D_i$. **a** Variation of $\Delta t_s$ (diamonds) and $\Delta t_r$ (circles), obtained from transient peaks in $D$ as indicated in the inset. Measurements were taken in 2 mM NaNO$_3$ solution and in response to positive – 0.2 V → + 0.2 V (green) and negative + 0.2 V → – 0.2 V (red) step potentials. The black dashed line and the black solid line are, respectively, the calculated values of $\Delta t_s$ and $\Delta t_r$ at different $D_i$ values based on equation (3). The solid red line describes the relaxation time $\Delta t_r$ in the presence of hydrodynamic damping alone as derived from calculated curves (Fig. 3a) at different $D_i$ values using equation (2'), as shown in Fig. 3a for selected peaks. **b** EDL charging time $\tau_{EDL}$ ($\Delta t_s + \Delta t_r$), at different surface separations $D_i$ in response to positive $-0.2$ V → $+ 0.2$ V (green squares) and negative $+ 0.2$ V → $-0.2$ V (red squares) step potentials in 2 mM NaNO$_3$ (taken from data in (**a**)); and in 5 mM NaNO$_3$ (black triangles). The upper and lower black solid lines are the calculated values of ($\Delta t_s + \Delta t_r$) at different $D_i$ values based on equation (3), for the 2 mM and 5 mM cases, respectively, where all parameters are fully determined (no adjustable parameters) (Supplementary note 6). The solid blue curve is the best fit of the characteristic time according to the transmission line model, $\tau_{TL} = (L^2 \lambda_D / \mathcal{D} h_p)$, using a Debye length of $\lambda_D = 6.8$ nm, diffusion coefficient of a nitrate ion $\mathcal{D} = 1.6 \times 10^{-9}$ m$^2$ s$^{-1}$ and $h_p = D_i$. The fit to the data shown corresponds to $L = 150$ μm. **c** The EDL charging time ($\Delta t_s + \Delta t_r$), at different surface separations $D_i$ in response to positive – 0.1 V → + 0.2 V (green squares) and negative + 0.2 V → –0.1 V (red squares) step potentials in 2 mM LiClO$_4$

the dynamics (see Figs. 3a and red curve in 4a); indeed, the difference between the charging times for anionic vs. cationic transport demonstrates clearly that the decay of the transient forces that we measure is dictated by diffusive-transport of ions from the reservoir. Finally, we would expect that different potential steps would result in a different charging times through their effect on the surface charge density change $\Delta\sigma$ and hence on the effective slit radius $L$ ($\sim (\Delta\sigma)^{1/2}$), as indeed is seen (Supplementary Figure 2). Thus, this broad agreement of the absolute magnitude as well as the variation of the predicted charging times $\tau_{TL} = (L^2 / \mathcal{D})(\lambda_D / h_p)$ with our measured EDL charging for different slit widths, salt concentrations, ionic mobility and surface potential changes, indicates the validity of the TL concept and its scaled behavior for our system.

## Discussion

In summary, we report the first measurements of EDL charging dynamics within a single nano-confined circular pore (or slit) following a step change in the surface potential (and charge) of one of the confining surfaces. This is done, using an SFB, by monitoring the transient change in surface forces due to an instantaneous electric field arising from the EDL perturbation, which decays as the EDL charges. Our results are in line with the characteristic times expected from the transmission line model applied to our nano-slit geometry, and indicate that such an approach could be used for probing EDL charging dynamics within single nano-slits in different conditions and aqueous environments.

## Methods

**Materials.** Gold pellets, 99.999% pure, were purchased from Kurt J. Lesker and evaporated from a graphite crucible. Sodium nitrate, NaNO$_3$, 99.99% pure was purchased from Merck Millipore and used as received. Lithium perchlorate, LiClO$_4$, 99.99% pure was purchased from Sigma-Aldrich and used as received.

**Preparation of salt solutions.** NaNO$_3$ and LiClO$_4$ were dissolved in purified water with a total organic content of less than 1 ppb (TOC < 1), a resistivity of 18.2 MΩ cm and pH = 5.8.

$1.8 \times 10^{-9}$ m$^2$ s$^{-1}$ for the Li$^+$ and ClO$_4^-$ ions respectively) differ by twice as much as for the former, there is a more marked effect of the mobility of the ion type permeating the slit on the charging time. This is seen by the differences between the red (charging by Li$^+$ transport) and green (charging by ClO$_4^-$ transport) data in Fig. 4c. This is especially the case in Fig. 4c for $D_i >$ ca. 80 nm, where ion transport rather than hydrodynamic effects dominates

**Surface force balance experiment**. An atomically smooth back silvered mica and molecularly smooth gold surface are glued to cylindrical fused-silica lenses (curvature radius $R \approx 1$ cm) in a crossed-cylinder configuration, equivalent to the geometry of a sphere on a flat, with the lower lens (mica) mounted on a horizontal leaf spring (spring constant $K = 81.5 \pm 2.7$ Nm$^{-1}$) and the top lens (gold) mounted on a sectored piezoelectric tube (PZT). The gold surface, serving as a working electrode (W), is connected to a potentiostat together with two platinum wires, acting as reference (R) and counter (C) electrodes, as described in detail elsewhere[44,45]. During all measurements, the electrodes are immersed in an electrolyte solution inside a quartz bath, custom-designed to prevent leakage of current to the ground, and the distance $H$ between them is of order a few mm. The distance between the surfaces is measured with a resolution of $1.0 \pm 0.3$ nm using multiple beam optical interferometry, and the normal forces between the surfaces are monitored through the bending of the horizontal spring and measured via a dynamic approach with a force sensitivity of ~50–100 nN, as previously described in detail[48,49].

**Video capture and analysis**. The absolute surface separation between the surfaces during EDL charging and dynamic force measurements was continuously monitored, by capturing the fringes of equal chromatic order (FECO) position, using a fast video recording camera (SONY XC-HR70) with a frame rate of 60 frames per s. In our gold/medium/mica/silver two-layer interferometer system each pixel is equivalent to an absolute distance of ca. 0.2–0.3 nm. The separation between the surfaces was determined using the multilayer matrix method, as previously described in detail[50,51].

**Data availability:**. The data sets generated and analyzed during the current study are available from the corresponding author on reasonable request.

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

## Acknowledgements

We thank Ohad Cohen for his help with the numerical solutions, Alexander Vaskevich for helpful discussions and Maarten Biesheuvel for useful comments. We thank the European Research Council (Advanced grant CartiLube), the Israel Science Foundation and the Minerva Foundation for financial support. This research was made possible in part by the historic generosity of the Harold Perlman family.

## Author contribution

J.K., R.T. and G.S. conceived the idea and designed the experiments. R.T. conducted the experiments and analyzed the data. S.S. and P.P. assisted with theoretical aspects of EDL charging dynamics. R.T. and J.K. wrote the manuscript. All authors discussed the results and commented on the manuscript.

## Additional information

**Competing interests:** The authors declare no competing interests.

