## [Peer Review file · Nature Communications]

Reviewers' comments:

Reviewer #1 (Remarks to the Author):

review of Tivony

The paper is highly relevant, and fitting with the prestige of this journal. This paper reports for the first time on experimental observations of charging times in a single nanopore. Up till now, this was only probed indirectly in macroscopic studies of complete electrodes. Thanks to this novel line of approach, a more in-depth and scientific study of the capacitive charging of porous electrodes becomes possible. The authors correctly point out in the introduction the wide range of important applications of these systems, and thus increased knowledge here is of societal importance.

It is a really interesting experiment, where abruptly a macroscopic (between the gold and mica) electric field is applied, which can be measured by the cantilever deflection, then relaxes when the ions redistribute in the EDL.

The paper presents its results well, and theory and data are in good agreement.

The paper can be published after minor revisions

There are a few things to improve.

1. Don't we need more information on the location of the reference and counter-electrode? Their position must play a role in the final experimental result of the 0.5 sec charging time ? The authors write that the charging time relates to ion currents from outside to into the slit, i.e., from bulk. So does the exact position of the ref and counter electrode matter? This is not a classical experiment with bulk solutions and 1 M salt !

2. page 4. I also wonder how fast is the "abrupt" potential change applied to the gold? in the context of this study, it should really be milli-secs or less to be called abrupt ?!

3. An equation like Eq. 1, I don't see any references about it. Also its typographical representation is not optimal, can be improved ! Same for other equations.

Small remarks:

Both in abstract and introduction "mesoporous" is referred to as < 1 micron. I am fine if the authors want to stick to it, but many a reader will object because IUPAC defines mesopores as having a size between 2 and 50 nm, with macropores having a size >50 nm.

The authors use the term "capacitive desalination." However, this water desalination technology is

called “capacitive deionization”. This is the name that is used by scientists working in this class of technologies. This name originates from LLNL (Livermore, California), and was first used in 1995. Please just check the prevalence of “capacitive deionization” vs “capacitive desalination” in google scholar.

page 1 “increasing at smaller slit thickness” is unclear, because is the “charging [speed]” increasing, or is the time increasing (making things go slower) ? often things go faster when distances get smaller, so therefore the doubt. and also for “decreasing with ion concentration”, so the question, what exactly is in/decreasing?

page 2: not K_B for Boltzmann’s constant, but k_B

Here in this equation “with c_0 the concentration of ions”, is not correct, is the concentration of salt (molecules). The concentration of all ions, is twice higher !

Reviewer #2 (Remarks to the Author):

The authors report experimental study of charging single nanopores using SFB technique. The idea is that, after the abrupt application of a potential, the charging of pore formed by metallic surface modulates the interactions between the surfaces, and such modulation can be used to gain insights into the charging dynamics of nanopores. Some analysis has been performed to confirm that the observed distance vs. time evolution is not an artifact due to hydrodynamic relaxation, and the results on charging dynamics are in qualitative agreement with expectations from the classical transmission line models.

The novelty of the work lies in the application of SFB for studying charging dynamics at single nanopore level. The analysis is generally competent and carefully done. For these reasons, I support the publication of this paper. However, there are several issues need to be addressed before this work can be accepted.

First, the authors claimed that charging of planar electrodes in bulk solution is characterized by a time scale $t_1 = \lambda_D^2/D$. This is not correct in this reviewer’s opinion. Charging of a planar electrode in a bulk solution is not a well-defined problem – one must ask where is the counter-electrode in such a problem? In fact, once a counter-electrode is defined, the characteristic distance between the working and counter- electrodes, L , will control the charging dynamics. In fact, one can define a $t_2 = L^2/D$. The correct characteristic time scale for charging in this case is $t = \sqrt{t_1*t_2}$. This has been explained in Bazant’s PRE paper. The t_1 the author talked about is the relaxation time for EDL. It is an intrinsic property of the EDL rather than a measurement of the time scale for charging following an impulsively imposed potential difference.

Second, the authors attribute the sharp rise of Fe after the application of a potential difference to the electrification of the metallic surface and the decay of Fe to the screening of the charge on the metallic surface by the influx of counter-ions into the nanopore. While this is reasonable idea, I would like to see

more quantitative analysis to support this. Specifically, after a potential difference is applied, the metallic surface is far from uniformly charged – only the part close to the pore exit acquires large charge, and those in the pore interior are barely charged. As time increases, the interior of the pore becomes charged. If this is the case, the working principle of measuring charging dynamics argued by the authors will be significantly modified. The authors should perform a simulation of the charging of a nanopore using the PNP model to clarify how the charge on the electrode surface evolves and thus put their interpretation on a firmer ground.

Some minor issues:

Page 2, “the ion distribution of the EDL at a charged surface screens ...”: how can “ion distribution” screen electric field?

Page 10 (line 1-2), comment on the origins of the factor of 2 difference.

Reviewers' comments:

Reviewer #1 (Remarks to the Author):

review of Tivony

The paper is highly relevant, and fitting with the prestige of this journal. This paper reports for the first time on experimental observations of charging times in a single nanopore. Up till now, this was only probed indirectly in macroscopic studies of complete electrodes. Thanks to this novel line of approach, a more in-depth and scientific study of the capacitive charging of porous electrodes becomes possible. The authors correctly point out in the introduction the wide range of important applications of these systems, and thus increased knowledge here is of societal importance.

It is a really interesting experiment, where abruptly a macroscopic (between the gold and mica) electric field is applied, which can be measured by the cantilever deflection, then relaxes when the ions redistribute in the EDL.

The paper presents its results well, and theory and data are in good agreement.

The paper can be published after minor revisions

We appreciate this concise and correct summary of our paper and thank the reviewer for these kind statements.

There are a few things to improve.

1. Don't we need more information on the location of the reference and counter-electrode? Their position must play a role in the final experimental result of the 0.5 sec charging time ? The authors write that the charging time relates to ion currents from outside to into the slit, i.e., from bulk. So does the exact position of the ref and counter electrode matter? This is not a classical experiment with bulk solutions and 1 M salt !

This is a perceptive comment and we now address this in the revised ms. as indicated further below (blue text).

The three-electrode electrochemical cell setup described in our ms. consists of two platinum electrodes (reference and counter) and a gold electrode (working) immersed in an electrolyte solution, where the distance between all electrodes is of order a few mm. In this configuration, as in conventional electrochemical cells, the actual voltage between the reference and working electrodes is affected by their relative position due to ohmic drop. In addition, the characteristic charging time of an ideally polarized electrode is $\tau_c = \sqrt{\tau_D \tau_b} = \lambda_D H / D$ (Bazant *et al.*, *Physical review E*, 2004), where $\tau_D = \lambda_D^2 / D$ and $\tau_b = H^2 / D$ are an intrinsic EDL charging time and a bulk diffusion time, respectively (λ_D is the Debye length, H is the separation between the electrodes and D is the ion diffusion coefficient). Therefore, as correctly commented by this reviewer, the process of EDL charging at surfaces (outside the nanopore) is indeed affected by the relative position of the electrodes.

However, that for the electrode separation of $H =$ say 5 mm and salt concentration of 1mM (i.e. $\lambda_D = 9.6nm$) used in our experiments the predicted charging time of an ideal electrode, $\tau_c = 3 \times 10^{-2}$ sec, is ca. 1.5 orders of magnitude smaller than our measured charging times 0.5 – 1.5 sec for the confined nano-slit. Moreover, while τ_c is not sensitive to the degree of confinement (i.e. the separation between gold and mica), the EDL charging time inside the nano-pores in our experiment scales roughly inversely with pore width h_p . Thus, the relaxation times we measure in our experiments are not affected by τ_c , which is very much smaller (and thus are not sensitive to the electrode separation H), but rather reflect the charging times of the nanopores as we discuss.

To clarify, we now added the following to our revised ms.:

Abstract: “We find that EDL charging time within a confined nano-slit occurs over times of order 1 s (far slower than the time of order 3×10^{-2} sec characteristic of charging an unconfined surface in our configuration)”.

Main text (page 2, line 19): This is well understood for planar electrodes in bulk solution, where the characteristic charging time is defined by $\tau_c = \lambda_D H / D$ (where H is the distance between electrodes and D is the ion diffusion coefficient)¹⁹⁻²², but for the case of EDL charging in nano-confined geometries (as in porous electrodes) the dynamics may be much slower.”

Main text (page 3, line 24): Our results show that full EDL charging within the nano-slit occurs over times of order 0.5 - 1 s, far longer than $\tau_c \approx 3 \times 10^{-2}$ sec in our configuration (where $\tau_c = \lambda_D H / D$, and the values of λ_D , H and D are given below).

Materials and methods section (page 12, line 14): “During all measurements, the electrodes are immersed in an electrolyte solution inside a quartz bath, custom-designed to prevent leakage of current to the ground, and the distance H between them is of order a few mm.”

2. page 4. I also wonder how fast is the “abrupt” potential change applied to the gold? in the context of this study, it should really be milli-secs or less to be called abrupt ?!

Following a potential step, the gold acquires its new surface potential value very rapidly (much faster than EDL charging time) as it is associated with drift motion of electrons which is orders of magnitude faster than for ions in solution (Bard, A. J. and L. R. Faulkner, *Electrochemical methods: fundamentals and applications*, New York: Wiley, 2001). Therefore, in our paper the potential change is referred as “instantaneous” or “abrupt”.

To clarify this point we now add the following:

Page 3, line 20: “...followed by a slower process in which the ionic atmosphere of the EDL rebuilds to screen the generated electric field while charging the nano-pore interior (the gold acquires its new surface potential value over a time δt much faster than EDL charging time as it is associated with drift motion of electrons which is orders of magnitude faster than for ions in solution^{11,40}).”

3. An equation like Eq. 1, I don't see any references about it. Also its typographical representation is not optimal, can be improved ! Same for other equations.

Equation 1 describes the balance of forces acting on the mica surface (mounted on a spring) during its motion in response to a step potential and is thus specific to our system, essentially Newton's equation with the negligible inertial term omitted as described in the text. However, we note that all of the included terms in these equations (such as the hydrodynamic damping force and electrostatic force) are properly referenced.

All equations will be typographically adjusted (if necessary) to meet journal specifications.

Small remarks:

Both in abstract and introduction “mesoporous” is referred to as < 1 micron. I am fine if the authors want to stick to it, but many a reader will object because IUPAC defines mesopores as having a size between 2 and 50 nm, with macropores having a size >50 nm.

We thank the reviewer for his comment. However, we would like to stick with the term mesoporous since we have data on pores both smaller and larger than 50nm (typically between 25-400nm).

The authors use the term “capacitive desalination.” However, this water desalination technology is called “capacitive deionization”. This is the name that is used by scientists working in this class of technologies. This name originates from LLNL (Livermore, California), and was first used in 1995. Please just check the prevalence of “capacitive deionization” vs “capacitive desalination” in google scholar.

We agree and make the change:

Page 2, line 4: “...while emerging applications include capacitive **deionization**...”

page 1 “increasing at smaller slit thickness” is unclear, because is the “charging [speed]” increasing, or is the time increasing (making things go slower) ? often things go faster when distances get smaller, so therefore the doubt. and also for “decreasing with ion concentration”, so the question, what exactly is in/decreasing?

We thank the reviewer for pointing this out. In both cases it is the EDL charging time that is increasing/decreasing. We now clarify this:

Page 1, line 17: “We find that EDL charging **time** within a nano-slit occurs over times...”

page 2: not K_B for Boltzmann’s constant, but k_B

Now corrected.

Here in this equation “with c_0 the concentration of ions”, is not correct, is the concentration of salt (molecules). The concentration of all ions, is twice higher !

Now corrected as follows:

Page 2, line 15: “...where c_0 is the concentration of **salt** in the bulk solution...”

Reviewer #2 (Remarks to the Author):

The authors report experimental study of charging single nanopores using SFB technique. The idea is that, after the abrupt application of a potential, the charging of pore formed by metallic surface modulates the interactions between the surfaces, and such modulation can be used to gain insights into the charging dynamics of nanopores. Some analysis has been performed to confirm that the observed distance vs. time evolution is not an artifact due to hydrodynamic relaxation, and the results on charging dynamics are in qualitative agreement with expectations from the classical transmission line models.

We appreciate this succinct and accurate description of our paper.

The novelty of the work lies in the application of SFB for studying charging dynamics at single nanopore level. The analysis is generally competent and carefully done. For these reasons, I support the publication of this paper.

We thank the reviewer for these kind comments.

However, there are several issues need to be addressed before this work can be accepted.

First, the authors claimed that charging of planar electrodes in bulk solution is characterized by a time scale $t_1 = \lambda_D^2/D$. This is not correct in this reviewer's opinion. Charging of a planar electrode in a bulk solution is not a well-defined problem – one must ask where is the counter-electrode in such a problem? In fact, once a counter-electrode is defined, the characteristic distance between the working and counter-electrodes, L , will control the charging dynamics. In fact, one can define a $t_2 = L^2/D$. The correct characteristic time scale for charging in this case is $t = \sqrt{t_1 t_2}$. This has been explained in Bazant's PRE paper. The t_1 the author talked about is the relaxation time for EDL. It is an intrinsic property of the EDL rather than a measurement of the time scale for charging following an impulsively imposed potential difference.

We agree with this perceptive comment, which is also related to comment 1 of reviewer #1. We address this issue in detail in our response to the first comment by reviewer #1 above, as well as several associated revisions to the ms. as below:

Abstract: “We find that EDL charging time within a confined nano-slit occurs over times of order 1 s (far slower than the time of order 3×10^{-2} sec characteristic of charging an unconfined surface in our configuration)”.

Main text (Page 2, line 19): This is well understood for planar electrodes in bulk solution, where the characteristic charging time is defined by $\tau_c = \lambda_D H/D$ (where H is the distance between electrodes and D is the ion diffusion coefficient)¹⁹⁻²²...”.

Main text (page 3, line 24): “Our results show that full EDL charging within the nano-slit occurs over times of order 0.5 - 1 s, far longer than $\tau_c \approx 3 \times 10^{-2}$ sec in our configuration (where $\tau_c = \lambda_D H / \mathcal{D}$, and the values of λ_D , H and \mathcal{D} are given below).”

Materials and methods section (page 12, line 14): “During all measurements, the electrodes are immersed in an electrolyte solution inside a quartz bath, custom-designed to prevent leakage of current to the ground, and the distance H between them is of order a few mm.”

Second, the authors attribute the sharp rise of Fe after the application of a potential difference to the electrification of the metallic surface and the decay of Fe to the screening of the charge on the metallic surface by the influx of counter-ions into the nanopore. While this is reasonable idea, I would like to see more quantitative analysis to support this. Specifically, after a potential difference is applied, the metallic surface is far from uniformly charged – only the part close to the pore exit acquires large charge, and those in the pore interior are barely charged. As time increases, the interior of the pore becomes charged. If this is the case, the working principle of measuring charging dynamics argued by the authors will be significantly modified. The authors should perform a simulation of the charging of a nanopore using the PNP model to clarify how the charge on the electrode surface evolves and thus put their interpretation on a firmer ground.

This is indeed a salient point which, thanks to the reviewer’s comment, we now clarify in the revised paper (see blue text further below), and address below.

Just before the potential change is applied, the gold surface is in equilibrium with the salt solution, and the electrostatic double layer (EDL) has essentially the same ion distribution everywhere, both within the confined nanopore surface and away from the nanopore. This is because the closest gold-mica separation \mathcal{D} defining the pore width (which we vary in our experiments from ca. 25 - 400 nm) is always much larger than the Debye length (typically 5 nm). At $t = 0$ the step potential is applied, and within a short time δt (related to the plasma frequency within the metal, and typically $< 10^{-9}$ sec) metallic charge is quickly accumulated, distributed uniformly at the gold surface to a value σ_0 everywhere, and the gold surface attains its new potential. The uniformity of σ_0 at $t = \delta t$ both within the nanopore and away from it on the unconfined gold surface is ensured by the fact that δt is very much shorter than the time scales over which changes to the EDL occur, such as τ_c discussed above. This is because equilibration within the metal is due to electron motion which is far more rapid than ionic transport, so that the EDL, and thus σ_0 , is still the same everywhere at δt .

This very fast (or as we referred in our paper ‘instantaneous’) change of the gold surface potential generates (at δt) an unscreened electric field $F_e(t = \delta t \approx 0)$ between the gold and the mica (which exerts a force on the charged mica). Away from the nanopore – at the unconfined gold surface - the value of this field decays over a time τ_c as discussed earlier, as the EDL equilibrates to fully screen it, and the surface charge density changes

to σ_1 say. Within the nanopore the decay of the field, and that of the corresponding force on the mica (which is what we measure) is much slower than τ_c . This field decays gradually to zero as the EDL at the gold surface within in the nanopore reverts – as counterions enter the pore - to its new equilibrium value in response to the potential jump, fully screening the field. Over this longer time the gold surface charge density within the pore will also change to σ_1 , as pointed out by the reviewer.

However, the key point (which is now clarified in the revised manuscript) is that the overall decay of the initial, unscreened electric field (at $t = \delta t$), and thus of the force on the mica, through the combination of screening and surface charge redistribution (from σ_0 to σ_1) on the nanopore gold surface, is associated with the equilibration of the EDL within the pore. It is this overall time which is measured and discussed in the ms.

We have now made the following changes to clarify this:

Page 3, line 16: "...in response to a step-change in the surface potential (or charge) of one of them..."

Page 3, line 20: "...followed by a slower process in which the ionic atmosphere of the EDL rebuilds to screen the generated electric field while charging the nano-pore interior (the gold acquires its new surface potential value over a time δt much faster than EDL charging time as it is associated with drift motion of electrons which is orders of magnitude faster than for ions in solution^{11,40})."

Page 4 line 20: "Then, either a positive ($-0.2V \rightarrow +0.2V$) or negative ($+0.2V \rightarrow -0.2V$) potential step $\Delta\Psi_{app}$ is applied to the gold while monitoring the gold-mica separation D (fig. 1C). After $\Delta\Psi_{app}$ is applied at time $t=0$ (fig. 1D), the gold electrode – both outside and within the confined nano-slit – accumulates a new surface charge density over a very short time δt (typically $< 10^{-9}$ sec due to drift motion of electrons^{11,40}) as it attains its new surface potential value Ψ_{gold} . This generates (at δt) an unscreened electric field which exerts an electrostatic force $F_e(t = \delta t \approx 0)$ on the negatively charged mica, bending the spring K_n on which it is mounted by ΔD to an extremal separation D_f relative to the gold, over a time Δt_s . As ions transport to reconstruct the EDL (EDL charging) within the nanopore, F_e is progressively screened and the surfaces relax to their initial separation D_i over a further time Δt_r (fig. 1D), since, in all cases in our study, $D_i \gg \lambda_D$. This transient motion manifests as a peak, with a characteristic asymmetric shape, in the D vs. t trace (fig. 1D) and a period of $(\Delta t_s + \Delta t_r)$ which characterizes the charging time of the EDL within the nano-slit. Away from the nano-slit – at the unconfined gold surface - the EDL equilibrates over a time τ_c ¹⁹ ($= \lambda_D H/D \approx 3 \times 10^{-2}$ sec), which is much shorter than the EDL charging time (0.5-2sec) within the nanopore measured in our experiments. For example, for the configuration of fig. 1D, a characteristic time $(\Delta t_s + \Delta t_r) \approx 0.85$ sec was obtained in response to a potential step of $-0.2V \rightarrow +0.2V$. Overall, this indicates three different time scales following the potential step: $(\Delta t_s + \Delta t_r) \gg \tau_c \gg \delta t$. Thus, the

times ($\Delta t_s + \Delta t_r$) we measure in our experiments, corresponding to the EDL equilibration within the nano-pore, are well separated from τ_c , the time for EDL equilibration at the unconfined gold away from the nanopore.”

Page 7, line 14: “The decay with time of $F_e(t)$ is complex, depending on flux of ions between the bulk and the nano-slit which in turn modifies the initial gold surface charge (i.e. at $t=\delta t$) within the confined area until the EDL is fully relaxed.”

Concerning the suggestion to perform a simulation of the charging of a nanopore using the PNP model, we believe this could provide interesting information, but it would be a substantial undertaking which we think is out of the scope of our experimental study.

Some minor issues:

Page 2, “the ion distribution of the EDL at a charged surface screens ...”: how can “ion distribution” screen electric field?

We now change this to:

Page 2, line 14: “In equilibrium, the EDL at a charged surface screens...”

Page 10 (line 1-2), comment on the origins of the factor of 2 difference.

We attribute the factor 2 difference between the transmission line fit and our estimated L value mainly to the different geometries considered in both cases. The transmission line model considers a cylindrical nano-pore of length L and uniform thickness h_p , closed at one end and exposed to the bulk solution (i.e. a reservoir of ions) at its open end while the pore considered in our paper is more resembled to a disk-like slit opened at its two ends.

We now add the following comment to our paper:

Page 10, line 5: “This is within a factor 2 of the value $L = (2R\Delta\sigma/c_0e)^{1/2} \approx 87 \mu\text{m}$ evaluated for the corresponding potential step ($\Delta\Psi_{\text{app}} = +0.2\text{V} \rightarrow -0.2\text{V}$ in 2 mM salt) (SI), which is believed to arise from the different geometries considered in the two cases.

With this we have addressed all points raised by the reviewers, and thank them again for their thoughtful comments.

REVIEWERS' COMMENTS:

Reviewer #2 (Remarks to the Author):

I am largely satisfied by the revisions made by the authors and support the publication of the manuscript.